# ABO Blood Types and COVID-19: Spurious, Anecdotal, or Truly Important Relationships? A Reasoned Review of Available Data

**DOI:** 10.3390/v13020160

**Published:** 2021-01-22

**Authors:** Jacques Le Pendu, Adrien Breiman, Jézabel Rocher, Michel Dion, Nathalie Ruvoën-Clouet

**Affiliations:** 1CRCINA, INSERM, Université de Nantes, F-44000 Nantes, France; Adrien.Breiman@univ-nantes.fr (A.B.); jezabel.rocher@univ-nantes.fr (J.R.); Nathalie.Ruvoen@univ-nantes.fr (N.R.-C.); 2CHU de Nantes, F-44000 Nantes, France; 3Microbiotes Hosts Antibiotics and Bacterial Resistances (MiHAR), Université de Nantes, F-44000 Nantes, France; michel.dion@univ-nantes.fr; 4Oniris, Ecole Nationale Vétérinaire, Agroalimentaire et de l’Alimentation, F-44307 Nantes, France

**Keywords:** COVID-19, ABO blood groups, natural antibodies, thrombosis, attack rate, susceptibility

## Abstract

Since the emergence of COVID-19, many publications have reported associations with ABO blood types. Despite between-study discrepancies, an overall consensus has emerged whereby blood group O appears associated with a lower risk of COVID-19, while non-O blood types appear detrimental. Two major hypotheses may explain these findings: First, natural anti-A and anti-B antibodies could be partially protective against SARS-CoV-2 virions carrying blood group antigens originating from non-O individuals. Second, O individuals are less prone to thrombosis and vascular dysfunction than non-O individuals and therefore could be at a lesser risk in case of severe lung dysfunction. Here, we review the literature on the topic in light of these hypotheses. We find that between-study variation may be explained by differences in study settings and that both mechanisms are likely at play. Moreover, as frequencies of ABO phenotypes are highly variable between populations or geographical areas, the ABO coefficient of variation, rather than the frequency of each individual phenotype is expected to determine impact of the ABO system on virus transmission. Accordingly, the ABO coefficient of variation correlates with COVID-19 prevalence. Overall, despite modest apparent risk differences between ABO subtypes, the ABO blood group system might play a major role in the COVID-19 pandemic when considered at the population level.

## 1. Introduction

Since the first description of COVID-19 in Wuhan, China, the emergence of the new SARS-CoV-2 coronavirus led to a global public health crisis due to massive morbidity and a burden of mortality that rapidly overwhelmed health systems worldwide. However, the disease impact shows considerable variation between countries and geographic areas for reasons that are poorly understood. Besides differences in the economical, sociological, behavioral, and political response to the pandemic, genetic factors might also play a role in their own right. Thus, soon after the beginning of the pandemic a publication from Wuhan, China, reported a higher risk of infection for people of blood group A, and inversely a lower risk for people of blood group O [1]. Since then, associations with the ABO blood groups have been described in several additional publications from China as well as many other locations from Asia, the Middle East, Europe, and North America [2,3,4,5,6,7,8,9,10,11,12,13,14,15,16,17,18,19,20,21,22,23,24,25,26,27,28,29,30,31,32,33,34,35,36]. Associations between ABO phenotypes were described with either the risk of infection or disease severity, although most studies did not explicitly separate these two aspects. Moreover, the ABO phenotypes that appeared associated with either a higher or a lower risk were not always identical across studies, and several studies failed to uncover any significant association [37,38,39,40,41]. Finally, if there is an effect of ABO blood groups on SARS-CoV-2 infection or the outcome of COVID-19, its impact on the evolution of the pandemic in different regions of the world and its potential importance in terms of patients care remain to be assessed. The purpose of this work is therefore to provide a reasoned review of the literature in order to assess the extent of the potential role and usefulness of the ABO blood group system in the COVID-19 pandemic.

## 2. The ABO Blood Group System at a Glance

The ABO blood group system was discovered more than a century ago and its understanding allowed the development of blood transfusion. However, the corresponding antigens are expressed on a large number of cell types in addition to erythrocytes [42]. They are of the carbohydrate type, constituting terminal motifs of either N-linked or O-linked chains of glycoproteins as well as of glycolipids. Their synthesis proceeds by addition of monosaccharide units to precursor glycan chains through specific glycosyltransferases. It first requires the synthesis of the histo-blood group H precursor antigen, which is catalyzed by alpha1,2fucosyltransferases that add a fucose in α1,2 linkage to a terminal β-galactose of the subjacent glycan chain. The FUT1 enzymes are responsible for this activity in erythroblasts, megakaryocytes, vascular endothelial cells, and several other cell types, while it is the FUT2 enzyme that catalyzes synthesis of the H antigen in most epithelial cells such as the upper airways, the digestive tract, and the lower genito-urinary tracts. Once the H antigen is produced, addition in α1,3 linkage of an N-acetylgalactosamine or of a galactose to the same subjacent galactose unit by the A or B blood group enzymes generates the A and B antigens, respectively (Figure 1). The A and B enzymes are coded by distinct alleles of the *ABO* gene, whereas the *O* alleles correspond to null alleles unable to generate any active enzyme. These three major types of alleles generate the four major phenotypes O, A, B, and AB [43]. Both the *FUT1* and *FUT2* genes also present null alleles that lead to a lack of precursor H antigen synthesis in the corresponding cell types and therefore to a lack of A and B blood group antigens expression in these cells [44]. *FUT1* null alleles are responsible for a rare red cell phenotype called “Bombay”. Given its rare occurrence, it will not be discussed any further. By contrast, null alleles of the *FUT2* gene are common and their frequency varies across populations. These alleles are responsible for the so-called “nonsecretor” phenotype which by contrast with the “secretor” phenotype is characterized by a lack of A, B, and H antigens in many secretions such as saliva and in epithelia. In the Western world, the secretors represent around 80% of the population and nonsecretors, the remaining 20% [44].

In addition to its antigens, the ABO system is characterized by the presence of antibodies against the A and B antigens. Thus, blood group O individuals possess anti-A and anti-B antibodies, blood group A individuals possess anti-B antibodies, and blood group B individuals have anti-A antibodies. Only blood group AB individuals are devoid of both anti-A and anti-B antibodies. This system of antigens and their cognate antibodies defines the basic rules of transfusion where blood group O constitutes a universal donor, whereas blood group AB represents a universal receiver [45]. The origin of the natural anti-ABO antibodies is still debated. Nonetheless, it seems that most of these antibodies appear during the first year of life under stimulation of microorganisms either pathogenic or from the microbiota that carry similar antigens [46,47]. Their amounts are highly variable between individuals and some data suggest that they may decrease with improved hygiene conditions [48,49].

The *ABO* gene and the *FUT2* gene, which controls expression of ABH antigens in epithelia, are among the few human genes clearly under frequency-dependent balanced selection, suggesting important roles in interactions with environmental factors [50,51,52,53,54,55]. Histo-blood group antigens, including ABO blood groups, have previously been implicated in the genetic susceptibility to several infectious diseases, including viral diseases. This has been particularly well documented for human noroviruses and rotaviruses that together are responsible for the majority of gastroenteritis cases worldwide. These non-enveloped RNA viruses attach to the carbohydrate antigens expressed in the gastrointestinal mucosa. They have evolved so that distinct strains recognize preferentially different carbohydrate motifs, resulting in a strain-dependent susceptibility in accordance with the person’s blood type [56]. Rabbit Hemorrhagic Disease Virus (RHDV) is a highly pathogenic rabbit calicivirus related to noroviruses also attaches to blood group antigens expressed in the rabbit respiratory and gut epithelia. The rabbit-RHDV pair allowed the documentation of a natural example of host–pathogen co-evolution involving the recognition of A, B, and H blood groups antigens by the virus [57]. These co-evolving host–pathogen pairs explain, at least partly, the maintenance of the ABO polymorphism. Nevertheless, using mathematical modeling, it has been argued that its maintenance would additionally require the role of the anti-ABO antibodies to be taken into account [58]. It has additionally been argued that viral-mediated selection may explain why South American native populations are exclusively of blood group O [59].

## 3. Hypotheses Linking ABO Types and COVID-19 and Their Consequences on the Interpretation of Reported Associations

Soon after the first report of an association between ABO phenotypes and the risk of COVID-19, several hypotheses were proposed about the underlying mechanisms [60,61,62,63]. Prior to presenting the results of the various studies reporting associations, or the lack of association, we will explain the bases and rationale of these hypotheses as they have distinct consequences that may explain discrepant outcomes stemming from variable designs of the association studies. Because of their crucial importance for blood transfusion ABO blood groups are generally perceived as red cell markers. Their potential role in COVID-19 is therefore not obvious. The mechanisms that have been proposed to account for their implication in the disease can be broadly divided into two major groups; those affecting the risk of infection and of SARS-CoV-2 transmission, on the one hand, and those affecting disease severity, on the other hand. 

### 3.1. The Anti-ABO Antibodies

As described above, histo-blood group antigens, including the ABH antigens, are synthesized by many epithelial cell types, including those of the respiratory and digestive tracts that are known to emit large amounts of viral particles [64,65,66,67]. The SARS-CoV-2 main envelope protein, the Spike or S protein harbors many glycosylation sites. It is therefore heavily glycosylated and structural analyses of the recombinant glycoprotein produced in HEK293 cells revealed a large panel of glycans, mostly of the N-glycan type, but also of O-glycans [68,69,70,71]. As the enzymatic machinery required for their synthesis is that of the host cell, the exact structures present on virions will depend on the infected cell type. Glycans detected on the recombinant S protein from HEK293 cells or on the virus produced in Vero cells therefore do not fully represent those that are synthesized on viral particles emitted from either respiratory or digestive epithelial cells from infected persons. Most importantly they cannot be decorated by ABH antigens as HEK293 cells and Vero cells do not synthesize these epitopes [72]. To uncover whether the S glycoprotein carries A, B, or H antigens when produced in cells that have the ability to synthesize these glycan motifs, in a recent work, we produced the recombinant S1 domain in CHO cells expressing the FUT2 enzyme and either the A or the B enzymes. This showed that H, A, or B epitopes can be present on the viral S protein in accordance with the glycosyltransferase repertoire of the cells [72]. It is therefore to be expected that authentic infectious virions produced by respiratory epithelial cells also carry the antigens in all individuals of the “secretor” phenotype. Consequently, applying the rules of blood transfusion, SARS-CoV-2 viral particles transmitted in ABO incompatible situations might be neutralized by the anti-A and anti-B antibodies. As blood group O individuals possess both types of antibodies, they might benefit from a better protection than blood group A or B individuals who possess only one of these types of antibodies and even more so than blood group AB people who have none of them. As previously discussed, this kind of protection will only be partial as it can only take place in incompatible ABO situations and likely requires sufficient amounts of the anti-A and anti-B antibodies [60]. Because these are highly variable, it follows that individuals who possess low titers of anti-A or of anti-B antibodies are expected to be at a higher risk of infection than people with high titers. This was indeed recently observed in a study where agglutination scores of patients were compared to those of a control group. Patients’ anti-A and/or anti-B scores were lower than those of non-COVID-19 controls. By contrast, no differences were observed for antibodies directed against similar carbohydrates, including the αGal antigen (see below), which cannot be synthesized by human cells, indicating that the lower scores of anti-ABO antibodies observed in patients did not result from a general decrease of anti-carbohydrate antibodies following infection [72]. 

Earlier observations additionally support the anti-ABO antibodies hypothesis. Following an outbreak of SARS-CoV in Hong Kong hospital in 2003, it was observed that blood group O hospital staff members in contact with the initial patient had been largely spared by the disease in comparison with non-O blood group staff members [73]. It could then be showed using an *in vitro* cellular model that the interaction between the virus S protein and the ACE2 protein, its receptor like that of SARS-CoV-2, could be specifically inhibited by anti-A antibodies when the S protein was produced by cells able to synthesize the blood group A antigen [74]. Additionally, Measles virus grown in cells expressing either the A or B blood group antigens was neutralized by human serum anti-B or anti-A antibodies, respectively, in a complement-dependent manner [75]. Likewise, the αGal xenoantigen, similar to the human B blood group antigen, but not present in humans due to mutations in the *GGTA1* gene that encodes a closely-related specific α1,3galactosyltransferase, is present in enveloped virus particles produced by animal cells possessing the enzyme [76]. A large body of evidence indicates that expression of the αGal carbohydrate epitope on viral envelopes leads to the elimination of viruses through anti-carbohydrate natural antibodies. Various mechanisms appear to be involved in the phenomenon, including blocking of receptor engagement, complement-dependent neutralization and amplification of the specific immune response through targeting of the opsonized virus particles to antigen presenting cells (reviewed in [77]). 

In light of all these data, the involvement of natural anti-ABO antibodies in COVID-19 infection is a serious possibility that needs careful attention. It should be stressed that if these antibodies play any role, they can only act by preventing infection or by decreasing the viral load. As soon as virus replication has taken place in the new host, newly formed virions will carry autologous glycans that can no longer be recognized by the allogeneic anti-ABO antibodies. Another critical consequence of a potential protection effect of anti-ABO antibodies is that, as protection exclusively takes place in situations of ABO incompatibility, the final number of infected individuals cannot be affected to a great extent, because more and more ABO compatible encounters will take place as the epidemic progresses. Yet, it can still have a major impact on the epidemic by significantly slowing down its progression which will enhance the efficacy of non-pharmaceutical measures of protection, such as social distancing. This was modeled using data from a SARS-CoV 2003 outbreak [74]. The model showed that in case of a strong protection afforded by a sufficient amount of anti-ABO antibodies in ABO incompatible transmission events, the epidemic would be very strongly delayed. 

### 3.2. The ABO Effect on Thrombosis 

A large number of studies show associations between ABO blood groups and thromboembolic diseases as recently reviewed [78]. This has been shown for myocardial infarction, atherosclerotic vascular disease, venous thromboembolism, and cardiovascular ischemic events. In all instances, people with non-O blood groups proved at a higher risk than O blood group individuals. One of the explanations rests on the observation that plasma levels of coagulation factors, most notably von Willebrand’s factor (vWF), are ~30% higher in non-O blood group individuals than in blood group O. Synthesis of vWF takes place in megakaryocytes and vascular endothelial cells that express ABH antigens. As it is heavily N- and O-glycosylated, it carries the antigens depending on the person’s ABO phenotype. Its clearance, largely glycan-driven and mediated by lectins such as the Ashwell hepatic lectin or CLEC4AM, is reduced in the presence of either the A, the B antigen, or both, leading to higher plasma levels of non-O individuals. In addition to their effect on hemostasis, there is evidence that ABO blood groups also affect vascular function, although the exact underlying mechanisms are not fully elucidated. In this context, it is noteworthy that the levels of vascular adhesion molecules such as the soluble forms of ICAM, P-selectin, and E-selectin correlate with ABO blood groups, with higher levels of these factors detected in blood group A individuals in comparison with blood group O [79]. Regardless of the precise underlying mechanisms, these observations suggest that ABO blood groups modulate leucocyte–endothelial interactions and influence the magnitude of the inflammatory response. 

Severe COVID-19 is characterized by an inflammatory state damaging the alveolar–capillary barrier and thereby compromising gas exchange. Intracapillary thrombosis and endothelial dysfunction are essential components of the severe form of the disease. Considering that ABO blood groups modulate both hemostasis and endothelial function, including its interactions with inflammatory cells, this has been suggested as an explanation of the reported associations between COVID-19 and ABO blood types [61,62]. Within the framework of this hypothesis, it appears that ABO blood groups would only influence the outcome of the disease at a late stage when ARDS or severe lung dysfunction has taken place. 

### 3.3. ABO Blood Groups and the Furin Cleavage Site 

Cell entry of SARS-CoV-2 involves pre-activation of the S protein by the proprotein convertase furin or furin-like proteases [80]. The furin cleavage site is surrounded by O-glycosylation sites, which represents a unique feature of SARS-CoV-2 among coronaviruses. As ABH antigens are largely present on O-glycans of epithelial cells [81], the distinct versions of O-glycans thus generated on the virions may impact the ability of furin to cleave the S protein. In a recent publication, Abdelmassih et al. also suggested potential relationships between ABO blood groups and furin [62]. They proposed that furin levels might be reduced in blood type O individuals based on a reported negative relationship between blood type O and furin-related proprotein convertases [82]. In addition, they suggested that furin levels modulated by the ABO phenotypes could play a role in the endothelial pathogenicity of SARS-CoV-2. In these conditions, the impact of ABO phenotypes on furin could take place both at the infection level and at the late stage of severe disease. 

### 3.4. ABO Blood Groups and Susceptibility to Other COVID-19-Associated Risk Factors 

Besides cardiovascular diseases, a large number of COVID-19 comorbidity factors have been described, some of which could also be associated with ABO blood groups. Although many studies looking for associations between ABO phenotypes and inflammatory conditions or autoimmune diseases have been conducted, they have often generated conflicting results. Nonetheless, a recent phenomic study involving a very large number of subjects from independent cohorts replicated some of these associations [83]. Thus, levels of C-reactive protein and alkaline phosphatase appear higher in blood group A individuals in comparison with blood group O, which may indicate a higher inflammatory state in the former group. Blood group O has also been reported to represent a protective factor for Crohn’s disease and ulcerative colitis, type I diabetes, and multiple sclerosis, although replications studies are lacking (reviewed in [62]). ABO blood groups additionally show associations with markers of the general metabolism. PSK9 levels appear higher in non-O blood group people, consistent with their higher levels of total cholesterol, LDL-C, and HDL-C [82,83]. Blood group A also appears associated with a lower forced vital capacity and forced expiratory volume in 1 s [83]. Although the mechanisms behind these various associations are unknown, they might contribute to increase the risk of severe COVID-19 or to worsen the disease. 

### 3.5. ABO Blood Groups and the Microbiota

As bacteria of the microbiota trigger the synthesis of anti-A and anti-B antibodies, composition of the gut microbiota might be critical to explain the large inter-individual variations in levels of these antibodies. The microbiota is also known to play a major role in controlling immunity and inflammation and since recent reports showed that the gut microbiota signature of COVID-19 patients appears distinct from that of healthy controls (reviewed in [84]), it is worth mentioning two studies that related ABO phenotypes to the composition of the gut microbiota. Mäkivuokko et al. observed that the overall microbiota composition of blood group B individuals differed from that of the other blood groups and showed increased levels of Actinobacteria in group A healthy individuals [85]. These bacteria are reportedly increased in Crohn’s disease and ulcerative colitis and may contribute to facilitate the development of the inflammatory state associated with severe COVID-19. Another study reported lower levels of Blautia in blood group A secretors than in non-A secretors [86]. A decreased level of these bacteria had previously been described in several inflammatory, autoimmune conditions, and aging [87], it may also reveal a higher risk of uncontrolled inflammation among blood group A COVID-19 patients than among other ABO blood types. 

Strikingly, all hypotheses linking ABO blood groups to COVID-19 predict a protective effect of the O blood type in comparison with non-O blood groups. Disentangling their relative importance is therefore not straightforward. Nonetheless, those that concern susceptibility to infection and the likelihood of transmission have different consequences than those predicting an impact on late events associated with the cytokine storm, ARDS and severe lung dysfunction. These differences help in analyzing the reported studies as discussed below. 

## 4. Studies Linking ABO Blood Types to COVID-19

### 4.1. Case–Control Studies Designed to Observe Associations 

At the time of writing, 34 case/control studies reporting associations between ABO phenotypes and the risk of COVID-19 have been reported, while only four studies failed to uncover a significant association (Table 1). Importantly, not all of the reports have been peer-reviewed as yet. As immediately apparent from Table 1, the data show a lower risk for O blood group people and/or inversely a higher risk for people with non-O blood types, most often blood group A. The reported significant odd ratios (ORs) generally revealed a modest influence of the ABO blood groups, although some reports suggested stronger effects. Thus, ORs for blood group O ranged from 0.53 to 0.9 and ORs for non-O blood groups ranged from 1.12 to 3.7. Meta-analyses of the earliest available data have already been conducted, confirming the effect [88,89,90,91,92].

The majority of these studies were hypothesis-driven. Yet, six of them concerned genome-wide association studies (GWAS) that looked for genetic associations without *a priori* [5,20,23,26,27,37]. Of the latter, only one failed to observe a significant signal at the *ABO* locus on the long arm of chromosome 9 (9q34) [37]. The fact that the majority of both agnostic GWAS and hypothesis-driven studies found the same significant associations with the ABO blood types indicates that these did not derive from biases due to the *a priori* search for a link between ABO blood groups and COVID-19. 

The various studies also differ in many other aspects, including the number of patients included, the definition of cases and the types of controls to whom patients were compared, as well as the relative ABO frequencies in the diverse studied populations. These different settings can seriously affect outcomes. Thus, in studies where the patients’ group was compared to blood donors, a bias might be introduced as blood group O is over-represented among regular blood donors because group O blood representing the universal donor is more widely demanded. In such studies, there is a risk of finding a false apparent increase in non-O blood types or a decrease of the O type among patients. Nonetheless, other studies using anthropological data of the frequency of ABO phenotypes obtained from large fractions of the local population alleviate this potential bias, as long as the population is sufficiently homogeneous and has not changed in composition since the acquisition of the anthropological data. In this respect, data from Asian and Middle-East countries are quite safe at the local level, with population ancestries being rather homogeneous. This is not so in the USA or European countries. Higher risks of COVID-19 have been reported for some disadvantaged subgroups from these countries [93]. The variations in ABO blood group frequencies between populations of different geographical origins and ancestries represent another important source of potential bias. In order to take this bias into account, in several studies, groups of patients and controls were stratified for ancestry [7,20,21,26,37]. Again, with the exception of one study discussed below [37], they consistently documented a lower risk of COVID-19 for blood group O, although not visible in all ethnic groups as illustrated by the study of Leaf et al. who uncovered a significant ABO effect in the white American population, but not in minorities of African Americans or Latin Americans [7]. This is likely due to the impact of the relative frequencies of ABO phenotypes in populations as discussed below. The effect of population admixture also likely explains why a study from New York, USA, failed to reveal any significant association between ABO blood types and the risk of COVID-19 [38]. In that study, no stratification for ancestry was performed even though the control group was composed of historical non-COVID-19 hospitalized patients. 

The number of patients included in the various studies has been highly variable, ranging from several thousands to less than one-hundred. Yet, they generated very similar results regardless of their high apparent differences in statistical power. This may be explained if the effect of ABO phenotypes was stronger in some geographical areas as discussed below. In that case, significant between-groups differences can be obtained with a lower number of cases. Inversely, in some geographical areas where the effect is weaker because of the relative frequencies of the ABO phenotypes, in particular a high frequency of blood group O, it takes very high numbers of patients to document a significant distortion of the ABO blood groups distribution in the patient groups, as illustrated by the lack of association reported from Brazil [41].

When only data on severe hospitalized patients are analyzed very important risk factors such as sex, age, obesity, cardiovascular, and metabolic underlying diseases are involved. As discussed above, the ABO polymorphism is known to associate with some of these COVID-19 risk factors, which might explain the link between blood group O and a lesser risk of COVID-19. However, analyses evaluating the potential confounding effect of these known risk factors showed that they cannot explain associations with ABO blood groups [2,12]. This conclusion is reinforced by a study from Iran where young patients (<45 y/o) without known underlying condition were enrolled. It showed a lower risk for blood group O in comparison with non-O types in absence of any strong comorbidity factor [13]. Inversely, one can also argue that the combined weight of these important risk factors is so high that it could mask any genetic effect on susceptibility to COVID-19 that would be more easily visible in the less severely affected patients who fortunately constitute the majority of patients. This could be the reason why one GWAS study failed to reveal a signal of association between COVID-19 and the ABO locus [37]. In that study, the control group was based on population data adjusted for ancestry, but the patient group was exclusively composed of patients under intensive care, excluding all non-severe cases and other hospitalized patients. Observing an effect on the risk of infection with this study design is therefore very difficult, unless the effect is very strong. Nonetheless, observing an effect on severity remains possible if a careful analysis of clinical data is performed within the patient group, which was not done in that particular study. In contrast, it was done in the study of Hoiland et al. [29] who also first compared ABO frequencies of patients requiring intensive care to those of the regional population and similarly failed to observe any difference between the patient and control groups. However, in a second analysis, even though it was based on a limited number of patients, the authors were able to uncover a clear effect of ABO phenotypes on severity by a careful analysis of ABO frequencies within the patient groups [29]. 

As the majority of COVID-19 infections result in a relatively mild disease, or even remain asymptomatic, studies taking into account all diagnosed patients (RT-PCR+), regardless of symptoms are well suited to find an effect on infection while inversely being unable to observe associations with severity. It is worth noting that such studies consistently observed either a lower risk for blood group O and/or an increased risk for non-O groups, in particular blood group A (Table 1). It therefore seems quite clear that blood group O individuals have a lower risk of SARS-CoV-2 infection relative to non-O blood types. The only exception comes from a study conducted in Bahrain where, in addition to the commonly observed increased risk for the A blood group, a significantly decreased risk was found for blood group AB [28]. It should nevertheless be kept in mind that the AB blood group is always the least represented and therefore that the data of that particular study rest on a rather limited number of cases.

### 4.2. Studies Designed to Observe an Effect on Severity

Considering the consistent associations found between the risk of COVID-19 infection and ABO blood types, observing potential associations with disease severity requires comparison of blood group frequencies between patient subgroups presenting with distinct clinical characteristics. As shown above, comparing patients to non-infected controls mainly reveals an effect on infection that will be diluted off when the disease has progressed to high severity. Revealing any effect on severity requires the comparison of blood group frequencies between patients suffering from more or less severe forms of the disease. Several studies of this type were performed and their design and results are briefly summarized in Table 2. Thus, Sardu et al. [30] reported that non-O Italian hypertensive COVID-19 patients had higher values of prothrombic indexes, cardiac injury, and rates of death than blood group O patients, consistent with the known effects of ABO blood groups on thrombosis and cardiovascular diseases. Likewise, a very careful clinical analysis of critically ill Canadian patients revealed that blood groups A and AB patients presented a higher risk of requiring mechanical ventilation, continuous renal replacement therapy, and prolonged admission to intensive care units than blood groups O and B patients [29]. In another study from Canada, the O blood group was also found associated with lower severity, whereas the B and AB blood groups were associated with higher severity [15]. In that study involving a large series of patients, the frequencies of ABO blood types were compared between cases with severe illness or death and all other cases, severe illness being defined as requirement for admission to intensive care unit. Independent data from a very interesting approach were reported from Spain. In their work, the authors compared hospitalized patients who required blood transfusion for their risk of death [12]. In this group of patients, blood group A and a non-O blood group were associated with a higher risk of death. Finally, a study on patients who had undergone transcatheter aortic valve replacement showed that having blood group A was the only independent predictor of COVID-19, but also predicted disease severity and death in this group of high-risk patients [31]. Therefore, although it appeared more difficult to evidence than its effect on infection, the association of blood group O with lower disease severity, or inversely the harmful consequence of having A or non-O blood groups, also appears now well established by these convergent studies, in accordance with the previously known effects of ABO blood types on thrombosis and the vascular function.

### 4.3. Other Study Designs

In addition to the studies presented in Table 1 and Table 2, several additional reports analyzed the risk of COVID-19 in relation to ABO blood types using quite distinct approaches. These are listed in Table 3. Thus, two studies compared the frequencies of ABO blood groups to COVID-19 prevalence between countries. The first one observed an association between the frequency of blood group A and prevalence of the disease, while the other observed an inverse relationship between the frequency of blood group B (including B + AB) and prevalence of the disease [32,33]. This approach is very interesting but suffers from two major weaknesses that will be discussed below. At this point, suffice it to say that they concern the low accuracy of prevalence data and, most critically, the fact that it is not the frequency of each blood type considered individually that matters at the population level.

Two other studies looked at the ABO types of blood samples collected for transfusion that proved SARS-CoV-2 seropositive. The selection of blood donors implies that all of these seropositive individuals had been free of symptoms during the two weeks preceding donation. In the study from France, these cases represented 3% of blood donations and presented an important deficit of blood group O in comparison with the remaining blood donors who did not have SARS-CoV-2-neutralizing antibodies [34]. Although the approach was similar, the study from Italy failed to find any bias in ABO blood types among the 1% of SARS-CoV-2 blood donors who had antibodies against the viral N protein [39]. Whether the discrepancy between the outcomes of these two studies is due to the different methods used to assess infection, the different percentage of seropositive blood samples, or the overall small number of concerned cases remains unclear.

Finally, another publication reported the analysis of a large outbreak that occurred on a French aircraft carrier [40]. While the vessel was in operation at sea, an outbreak occurred unabated for two weeks before confinement measures were taken. A large number of crew members were infected since the attack rate was 76%, but only 1.6% of the infected sailors could be considered as severe cases, that is, requiring oxygen therapy or admission to an intensive care unit. By comparing these 76% infected to the remaining 24%, no difference in ABO blood group frequencies could be observed. At first sight, this negative result appears to strongly contradict all the other reports discussed above that showed significant associations between the risk of COVID-19 and ABO phenotypes. Yet, when considered within the framework of the anti-ABO antibodies hypothesis, this result was predictable. Anti-ABO antibodies lower the risk of transmission at the group or population level, slowing progression of the epidemic [60,74]. In a confined situation such as that of an aircraft carrier where the outbreak lasted until the virus infected over three quarters of the personnel on board, nearly all susceptible individuals have been infected because of multiple contacts. It should be remembered that O blood group individuals are susceptible to transmission from other O blood group individuals. If the outbreak had been stopped early, then blood group O crew members might have proven less represented among the infected than among the non-infected crew members. 

## 5. Consequences of between-Populations Differences in ABO Blood Types Frequencies

Frequencies of the A, B, and O alleles and of the respective ABO phenotypes are quite variable across human populations [50]. Overall, on the Eurasian continent, there exists a gradient of blood group B whose frequency increases as one moves from west to east. This increase takes place at the expense of both blood groups A and O that are less frequent in East Asia than in Western Europe, the Middle East presenting intermediate frequencies [94]. The African continent is characterized by a rather large and complex diversity of ABO blood groups frequencies between geographic locations and ethnic groups [33]. The American continent is characterized by an extremely high frequency of the O blood type among Amerindian populations, most notable in its Central and Southern parts [94]. Considering these large differences between geographical areas and human populations, variation in the impact of ABO phenotypes on COVID-19 is to be expected. Documenting an effect is more difficult in ethnically mixed populations such as those of Western Europe or of the Americas. As already mentioned, special care should therefore be given to alleviate biases introduced by the population admixture as the risk of COVID-19 is clearly associated with socioeconomical conditions, themselves associated with the ethnic background. 

For all hypotheses raised to explain the links between ABO phenotypes and COVID-19, variations in relative frequencies represent a basic statistical issue. Obviously, it is more difficult to observe significant associations for less well represented phenotypes and this may contribute to explain some of the divergences observed between studies. In addition, and quite distinctly, within the framework of the anti-ABO hypothesis, the variation in ABO phenotypes frequencies affects the outcome in a more complex manner. Thus, anti-B antibodies from blood group A individuals may be protective from viral particles emitted by blood group B individuals. Reciprocally, anti-A antibodies from blood group B individuals can only be protective in the case of an ABO incompatible transmission event originating from a blood group A person [60]. The relative proportion of blood group A and B in the population will therefore be critical. This may explain why blood group A appears as a risk factor more often than blood group B. Indeed, many studies were performed on population showing a higher representation of blood group A than of blood group B. Likewise, O blood group individuals if protected from transmission from non-O individuals, cannot be protected from transmission by other O blood group individuals. Consequently, the frequency of the O blood type in a population will be critical. At the individual level, the maximal protection of O blood group individuals will take place when they only represent a small fraction of the population. Inversely, when present in a large proportion their protection by anti-ABO antibodies will vanish and completely disappear if all individuals of the population were of the O blood type. With this in mind, we looked for a correlation between the reported odd ratios of O versus non-O blood types to the frequencies of blood group O in the studied population listed in Table I. To do this, we used studies that reported the O blood group odd ratios in comparison with all other groups and we also used the frequency of blood group O in the studied populations as reported by the authors or deduced from previously published data. Figure 2A shows the results of the analysis, indicating that, as predicted by the hypothesis involving anti-ABO antibodies, O blood group individuals may benefit from a more important protection from SARS-CoV-2 infection in locations where blood group O frequencies are lower.

As the above observation suggested that the variable ABO frequencies across populations may differentially impact the outcome of the pandemic between countries, we looked for a potential correlation between ABO frequencies and SARS-CoV-2 attack rates worldwide. As already mentioned above, the relationship between an ABO blood type and susceptibility to COVID-19 is not straightforward, therefore individually relating each ABO blood type to the virus attack rate may be misleading. At the population level, the optimal protection that anti-ABO antibodies may provide should be attained when A, B, and O phenotypes are more or less evenly distributed because protection should occur only in ABO incompatible transmission events. In extreme situations, where one or two of these three major phenotypes largely dominate, incompatible events will be less frequently encountered. Thus, rather than looking for correlations between each ABO phenotype and SARS-CoV-2 attack rates, we looked for correlations between ABO coefficients of variation in different countries or geographical locations and SARS-CoV-2 attack rates. Protection conferred by anti-ABO antibodies should be optimal in populations presenting low coefficients of variation and inversely it is expected to be far less efficient or even not efficient at all in populations where they are elevated. ABO coefficients of variation were calculated using published frequencies of ABO blood types in different regions or countries. However, obtaining reliable SARS-CoV-2 attack rates that can be compared between countries is rather complex. Rates of detection obtained following RT-PCR testing are particularly unreliable since testing strategies have been very different between countries and within countries during the unfolding of the pandemic. Using data such as the number of COVID-19-related deaths/10^6^ inhabitants may appear easier to compare. Yet, these may be greatly affected by variable reporting strategies between countries and most importantly by the age structure of the population as the risk of death increases exponentially with age [95]. Older populations will therefore appear disproportionately affected, regardless of other parameters. More reliable data were recently provided by O’Driscoll et al. [95] using seroprevalence analyses from 45 countries or regions. Using these published median values of proportions of SARS-CoV-2 seropositive individuals as of September 2020, we found 41 countries with reliable published national ABO frequencies. As depicted on Figure 2B, a strong positive relationship can be detected between infection rates and the ABO coefficients of variation. It is noteworthy that countries of Central and South America where blood group O is highly prevalent show the highest SARS-CoV-2 attack rates, as predicted by the hypothesis on anti-ABO antibodies protection limited to ABO incompatible transmission events. Looking at a more homogeneous group of countries, both in terms of ABO coefficient of variation and of socioeconomic status, European countries also show a clear relationship between the distribution of ABO blood groups and infection attack rates (r^2^ = 0.29, *p* = 0.012).

Infection attack rates are very dependent on demographic and geographic variables, the national policies of protection that have been set up by governments, as well as by the socioeconomic conditions and inequalities [96,97,98]. To determine whether the observed associations between ABO coefficient of variation and SARS-CoV-2 attack rates actually revealed underlying covariation with historical between-population socioeconomic inequalities, we looked for a potential relationship between the ABO coefficient of variation and the inequality-adjusted human development index (IHDI). This index has been created by the United Nation Development Program (UNDP) in order to best define the degree of human development by taking inequalities into account. As shown on Figure 2C, we controlled that there was no relationship between ABO coefficient of variation and the IHDI. As unfavorable socioeconomic and demographic conditions have been discussed as important drivers of the epidemic [99], we also tested whether the IDHI was associated with the attack rates for the 41 countries that could be included in our analysis. This was indeed the case (Figure 2D), but the correlation was not as strong as that observed between attack rates and the ABO coefficients of variation. More refined indexes might be able to better capture the effect of inequalities and poverty on the impact of COVID-19, but that is beyond the scope of this article. Regardless, these observations suggest that the distribution of ABO blood groups strongly impacts development of the COVID-19 pandemic when looking at the level of populations. They suggest that the protective effect of anti-ABO antibodies is particularly high in Asian countries, important in Europe and North America, but unfortunately less important in countries of Central and South Americas where the frequency of blood group O is disproportionately high in comparison with those of blood groups A and B. This highly ABO-biased distribution is due to the large American Indian populations in these countries as American Indians are almost exclusively of blood group O [100]. Interestingly, in the USA, the impact of COVID-19 is extremely high among non-Hispanic American Indians and Alaska Natives with attack rates 3.5 times higher than in the white population [101]. It is also high among Hispanics (3 times higher compared to whites) [21], followed by African-American (2.3 times higher compared to white) in parallel with the frequencies of blood group O in these subgroups [21,102]. Similarly, a study from the UK aimed at relating ABO phenotypes to the risk of COVID-19 in pregnant mothers, reported the results of separate analyses of White and BAME (Black, Asian, Minority Ethnic) women, the latter being overrepresented among the COVID-19 positive cases [35]. The ABO coefficients of variation were widely different between the two subgroups, 60% and 28% for the White and BAME women groups, respectively. In accordance with the anti-ABO hypothesis, in the White COVID-19+ group, O blood group was modestly decreased in comparison to the White COVID-19− group (37% vs. 43%), but in the BAME cohort, the effect was much more pronounced (25% vs. 40%). It is thus tempting to hypothesize that the disproportionately high COVID-19 impact on the disadvantaged subgroups as well as in some countries might be explained by a synergistic effect between ABO blood group distribution and socioeconomic factors. 

Although correlative and potentially influenced by other demographic covariates that cannot be ruled out, the data presented in Figure 2 suggest that the protective effect of anti-ABO antibodies could greatly contribute to the large COVID-19 disparities between countries and population subgroups. It seems that non-pharmaceutical interventions, such as population confinement and social distancing, are efficient when the ABO coefficients of variation are low, as seen in some Asian countries that remarkably succeeded in controlling the epidemic. However, when they are high, the non-pharmaceutical interventions fail as illustrated by the examples of Latin American countries. When intermediate, as in Europe, they work to some extent, but are still affected by the West–East gradient of ABO frequencies, Eastern European countries tending to fare better, as of September 2020. All this seems to indicate that when the frequency of blood group O is below 40% and blood groups A and B are well balanced, the situation remains manageable, but when blood group O frequency increases above 40% and that the frequencies of blood groups A and B are unbalanced, management of the epidemic in absence of a vaccine turns out to be more difficult. 

## 6. Conclusions

Collectively, the data presented above indicate that ABO blood groups influence the risk of SARS-CoV-2 infection and several studies additionally allow to ascertain an effect of the ABO phenotypes on disease severity. In both situations, blood group O appears protective in comparison with non-O types. As discussed above, the protective effect on infection could be mediated either by natural anti-A and anti-B antibodies or by a lower efficiency of furin cleavage in blood group O individuals. Both could lead to either a complete protection or to lowering the initial viral load, which may have important consequences by facilitating viral clearance by the immune system and preventing the cytokine storm and ensuing ARDS. The observation of a link between ABO coefficient of variation in different geographical areas and the odds ratios of blood group O relative to other blood types as well as the COVID-19 attack rates suggest that anti-ABO antibodies play a prominent role in protection against infection but that their impact is heavily influenced by the relative frequencies of ABO phenotypes in the population. 

The previously well-documented influence of ABO phenotypes on hemostasis and vascular function, likely contributes to the lower severity of the disease observed among already severely affected patients of blood group O in comparison with non-O blood group patients. Other mechanisms could also contribute, including effects of the ABO polymorphism on inflammation and immune functions as well as on lipid metabolism. 

At the individual level, the increased risk of severe COVID-19 symptoms associated with non-O blood groups is generally modest. Therefore, it is not clear if that information can be useful at the clinical level. Nonetheless, in selected groups of patients, such as patients with underlying cardiovascular diseases it may be of importance, as illustrated by the study of Sardu et al. on hypertensive patients which detected a 3.7 times increased risk of death in non-O blood group patients [30]. Special attention could be important for such high-risk patients who may need early anticoagulant therapies in order to reduce cardiac injury, as suggested by the authors. 

Until recently, the only evidence that natural anti-ABO antibodies played any significant biological role, besides their importance for blood transfusion and organ transplantation, came from artificial *in vitro* observations and indirectly from the association between ABO phenotypes and SARS [74]. The new data accumulated on COVID-19 point to a role of these antibodies in the control of outbreaks at the population level. Anti-ABO antibodies could also help controlling infection by other coronaviruses as these viruses are highly glycosylated and replicate in epithelial cells of the upper respiratory tract that strongly express ABH antigens in accordance with the *ABO* and *FUT2* genes polymorphisms. There is evidence that anti-A and B titers are decreasing in relation with improved living conditions [49]. This phenomenon could contribute to facilitate virus transmission in contemporary societies with high standards of living. Moreover, COVID-19 patients have lower levels of anti-ABO antibodies than control individuals, suggesting that these antibodies are protective only when present in sufficient amounts [72]. Interestingly, levels of natural anti-glycan IgM, including anti-ABO, decrease with aging, which may contribute to the increased risk of infection in the elderly [103]. Acting to increase these antibodies titers at the population level would thus be desirable in order to take full profit of this natural antiviral defense mechanism. Although a detailed discussion of the means by which anti-ABO antibodies could be raised at the population level is beyond the scope of this article, a possibility would be to use selected probiotic bacterial strains that express the A and/or B antigens as it is largely under stimulation of bacteria from the microbiota that these antibodies are naturally raised. A transfusion accident was reported where a platelet donor had an extremely high anti-B titer following ingestion of a probiotic preparation containing several species of harmless bacteria [104]. Similarly, pediatric patients developed anti-B in association with probiotic use [105], suggesting that this could be a broadly applicable strategy as a complement to the SARS-CoV-2 vaccine strategy. Populations where ABO coefficients of variation are low or intermediate could slow down the virus transmission to a large extent, making it easier to control the epidemic. Even populations that present high ABO coefficients of variation could benefit from the anti-ABO protective effect, albeit to a lesser degree. That would increase the efficacy of both non-pharmaceutical interventions and of vaccines if used as a complementary tool in the fight against COVID-19. 

## Figures and Tables

**Figure 1 viruses-13-00160-f001:**
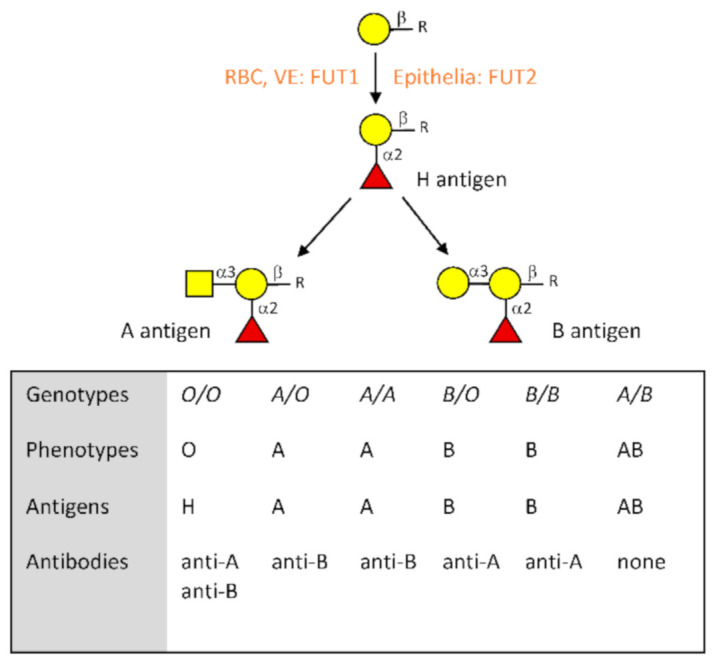
Description of the major characteristics of the ABO blood group system. Biosynthesis of the A and B antigens starts from a precursor structure constituted by a galactose residue in beta linkage to a subjacent sugar located at the termini of either N- or O-glycans as well as glycolipids. In red blood cells (RBC), vascular endothelial cells (VE) and other cell types such as megakaryocytes that give rise to platelets, addition of a fucose in α1,2 linkage by the FUT1 enzyme gives rise to the H blood group antigen. In most epithelial cells, synthesis of the H antigen is performed by the FUT2 enzyme. Blood group A antigen is then synthesized by the A enzyme coded by *A* alleles of the *ABO* gene, while blood group B antigen is synthesized by the B enzyme coded by *B* alleles. *O* alleles are unable to generate a functional enzyme; therefore, *O*/*O* individuals leave the H antigen unchanged. Relationships between genotypes, phenotypes, antigens, and the corresponding natural antibodies are shown.

**Figure 2 viruses-13-00160-f002:**
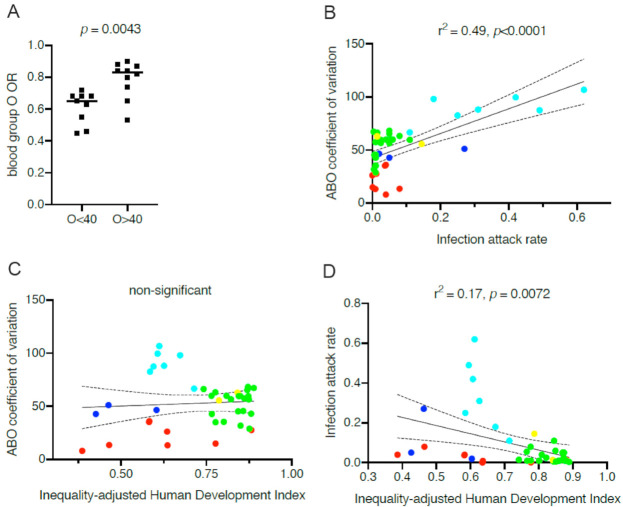
Relationship between ABO blood groups and the impact of COVID-19 across countries. (**A**) Blood group O odd ratios (ORs) of COVID-19 relative to the other blood groups and blood group O frequencies at various geographical locations. Odd ratios were obtained from the works in [1,2,3,6,9,10,11,14,17,18,19,22,24], and the frequencies of blood group O were those reported in the groups of controls of the corresponding studies, representing frequencies in the local populations. OR values in populations where frequencies of blood group O are below and above 40% (O < 0.4 and O > 0.4, respectively) were compared by a two-tailed Mann–Whitney test. (**B**) Regression analysis between coefficient of variation calculated from the distribution of ABO for each country and attack rates represented by median estimates calculated by O’Driscoll et al. [95] from seroprevalence studies. (**C**) Regression analysis between ABO coefficients of variation of each country and the 2018 IHDI (inequality-adjusted human development index) obtained from the UNDP (United Nation Development Program). (**D**) Regression analysis between COVID-19 attack rates of each country and the 2018 IHDI. Countries are labeled by geographic area. Cyan: Central and South Americas; blue: Africa; green: Europe; yellow: North America; red: Asia.

**Table 1 viruses-13-00160-t001:** Case–control studies reporting data on ABO blood groups and COVID-19.

Country	Cases		Controls	ABO Effect ^a^				Ref.
	Definition ^b^	Number	Definition ^c^	O	A	B	AB	
China	I	1775	I	<Risk	>Risk	ns	ns	[1]
USA	I	682	I	<Risk	>Risk	ns	ns	[2]
China	II	2153	I	<Risk	>Risk	ns	ns	[3]
Turkey	I	186	III	<Risk	>Risk	ns	ns	[4]
**Spain/Italy ^d^**	III	1610	III + IV	<Risk	>Risk	ns	ns	[5]
China	I	187	IV	<Risk	>Risk	ns	ns	[6]
USA	V	3239	V	<Risk	>Risk	ns	ns	[7]
China	III + IV	97	I	<Risk	>Risk	ns	ns	[8]
Turkey	I	179	IV	<Risk	>Risk	ns	ns	[9]
Iran	VII	76	I	<Risk	>Risk	ns	ns	[10]
China	V	134	I	<Risk	>Risk	ns	ns	[11]
Spain	VIII	854	VI	<Risk	>Risk	ns	ns	[12]
Spain	IX	965	III	<Risk	>Risk	ns	ns	[12]
UK	I	86	II	<Risk	>Risk	ns	ns	[35]
Turkey	II	1667	I	<Risk	>Risk	ns	ns	[36]
Iran	VI	93	I	<Risk	>Risk	>Risk	>Risk	[13]
USA	I	1289	II	<Risk	ns	>Risk	>Risk	[14]
Canada	I	7071	II	<Risk	ns	>Risk	>Risk	[15]
India	I	8102	I	<Risk	ns	>Risk	ns	[16]
Spain	II	226	I	<Risk	ns	>Risk	ns	[17]
Saudi-Arabia	II	72	III	<Risk	ns	ns	>Risk	[18]
Iran	I	397	IV	<Risk	ns	ns	>Risk	[19]
**USA + UK**	I	15,434	II	<Risk	ns	ns	ns	[20]
USA	I	34,178	II	<Risk	ns	ns	ns	[21]
Denmark	V	7422	I	<Risk	ns	ns	ns	[22]
**International**	I + II	6696 + 3199	II	<Risk	>Risk			[23]
Italy	I	447	III	<Risk	ns	ns	ns	[24]
China	II	103	II	ns	>Risk	ns	ns	[25]
**USA**	I	2417	I	ns	>Risk	ns	ns	[26]
**Italy/Spain**	III	505	VI	ns	>Risk	ns	>Risk	[27]
Bahrain	V	2334	III	ns	ns	>Risk	<Risk	[28]
Canada	V	95	I	ns	ns	ns	ns	[29]
**UK**	V	2244	I	ns	ns	ns	ns	[37]
Brazil	I	2037	II	ns	ns	ns	ns	[41]
USA	II	957	IV	ns	ns	ns	ns	[38]

^a^ All relevant studies in published or prepublication (not peer reviewed) formats as of 14 December 2020 are included. Significant associations are marked in green to indicate a decreased risk relative to the other blood types or in red to indicate an increased risk; ns: non-significant. ^b^ I: all diagnosed patients (RT-PCR), II: hospitalized patients, III: severely ill patients, IV: mildly ill patients, V: critically ill patients (ICU patients), VI: ICU admitted patients < 45 y/o with no previous history, VII: deceased patients, VIII: convalescent plasma donors (previously blood donors), IX: transfused patients. ^c^ I: regional anthropological data, II: regional SARS-CoV-2 RT-PCR negative people, III: blood transfusion donors, IV: non-COVID-19 hospitalized patients, V: non-COVID-19 hospitalized patients stratified by ethnic group, VI: first-time transfusion blood donors. ^d^ Genome-wide association studies (GWAS) studies are marked in bold.

**Table 2 viruses-13-00160-t002:** Studies specifically analyzing an effect on disease severity ^a^.

Country	Type of Study	ABO Effect ^b^				Ref.
		O	A	B	AB	
Italy	Analysis of clinical and biological criteria among hypertensive COVID-19 patients	<risk				[30]
Canada	Analysis of clinical and biological criteria among critically ill COVID-19 patients	<risk	>risk	<risk	>risk	[29]
Spain	Analysis of the risk of death among COVID-19 patients requiring transfusion	<risk		ns	ns	[12]
Canada	Separate analyses of patients with or without severe illness	<risk	ns	>risk	>risk	[15]
France	Comparison between patients who underwent transcatheter aortic valve replacement	ns	>risk	ns	ns	[31]

^a^ Studies including patients of various severity degrees, not specifying clinical criteria and comparing with non-COVID-19 control groups were not taken into account; ^b^ Significant associations are marked in green to indicate a decreased risk relative to the other blood types or in red to indicate an increased risk; ns: non-significant.

**Table 3 viruses-13-00160-t003:** Other study settings reporting data on ABO blood groups and COVID-19.

Country	Type of Study	ABO Effect ^a^				Ref.
		O	A	B	AB	
Europe, Africa, Middle-East, Asia	Association between frequency of the A allele and COVID-19 prevalence across countries		>Risk			[32]
Europe, Africa, Asia, America	Correlation between frequency of B+AB blood types and COVID-19 prevalence across countries			<Risk		[33]
France	Comparison of ABO phenotypes of blood donors with SARS-CoV-2 neutralizing antibodies	<Risk	>Risk	ns	ns	[34]
Italy	Comparison of ABO phenotypes of blood donors with SARS-CoV-2 anti-N antibodies	ns	ns	ns	ns	[39]
France	Comparison of ABO phenotypes of COVID+ vs. COVID− aircraft carrier crew members	ns	ns	ns	ns	[40]

^a^ Significant associations are marked in green to indicate a decreased risk relative to the other blood types or in red to indicate an increased risk; ns: non-significant.

## Data Availability

Data used to generate Figure 2 are available upoin request.

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
