# Peer review of "ABO Blood Types and COVID-19: Spurious, Anecdotal, or Truly Important Relationships? A Reasoned Review of Available Data"

_viruses, 2021, doi:10.3390/v13020160_

Round 1

Reviewer 1 Report

ABO Blood Types and COVID-19, Spurious, Anecdotical or 2 Truly Important Relationships? A Reasoned Review of Availa ble Data.  Jacques Le Pendu  , Adrien Breiman, Jézabel Rocher , Michel Dion , Nathalie Ruvoën-Clouet

This is an interesting comprehensive review on a subject which has been debated for the last year on whether there is a decreased infection by Covid 19 virus in blood group O individuals in comparison to that in individuals who are not of blood group O. Although people of blood group O can be infected by virus originating in blood group O individuals, their tendency to get infected by virus from blood group A individuals is significantly lower. The review discusses the possible role of natural anti-A antibodies in blood group O individuals against as well as other mechanisms (e.g., thrombosis) that may contribute to the increased resistance to infection and damage by Covid 19 virus originating in blood group A infected individuals. The updated information in this review is likely to be of interest to researchers studying Covid 19 infections and thus it warrants publication pending on few suggested minor revisions based on the following comments:

  1. In Table 1 the comments (a-d) should be explained.
  2. In regard to “…the maintenance of the ABO polymorphism” (line 120), The authors indicate that in South America there is large majority of blood group O individuals. it is important in this review to mention the exception of several South American Indian populations that are exclusively of blood group O and the possible relationship to viral mediated selection of this blood group. This issue is discussed in “Galili U. Host Synthesized Carbohydrate Antigens on Viral Glycoproteins as "Achilles' Heel" of Viruses Contributing to Anti-Viral Immune Protection. Int J Mol Sci. 2020 Sep 13;21(18):6702.
  3. The statement “ABH antigens are largely present on O-glycans of epithelial cells” should be based on a reference.
  4. “The American continent is characterized by an extremely high frequency of the O blood type among Amerindian populations” – The North American Indians (USA and Canada) have 30-40% blood group A. Thus, the term “Amerindians” requires definition.

Author Response

We thank the reviewer for his(her) interest and thoughtful comments (in italics below).

1. In Table 1 the comments (a-d) should be explained.

Explanations have been added to the corresponding comments. They unfortunately had been erased by the post-submission formatting of the manuscript made by the editorial office.

2. In regard to “…the maintenance of the ABO polymorphism” (line 120), The authors indicate that in South America there is large majority of blood group O individuals. it is important in this review to mention the exception of several South American Indian populations that are exclusively of blood group O and the possible relationship to viral mediated selection of this blood group. This issue is discussed in “Galili U. Host Synthesized Carbohydrate Antigens on Viral Glycoproteins as "Achilles' Heel" of Viruses Contributing to Anti-Viral Immune Protection. Int J Mol Sci. 2020 Sep 13;21(18):6702.

This has now been taken into account and the suggested reference has been added (ref #59).

3. The statement “ABH antigens are largely present on O-glycans of epithelial cells” should be based on a reference.

A specific reference has been added (Pelaseyed and Hansson 2020, ref #81)

4. “The American continent is characterized by an extremely high frequency of the O blood type among Amerindian populations” – The North American Indians (USA and Canada) have 30-40% blood group A. Thus, the term “Amerindians” requires definition.

We agree with the reviewer’s remark. Accordingly, the corresponding sentence has now been modified as follows: “The American continent is characterized by an extremely high frequency of the O blood type among Amerindian populations, most notable in its Central and Southern parts”. It should be kept in mind that even in North America, Native populations have an unusually high frequency of blood group O. The data on Covid-19 that we mentioned from Hatcher et al (ref #101) reports a 54.5% blood group O based on data from 21 states of the USA, which is very high in comparison with populations from Asia and even from Europe.

Reviewer 2 Report

A very detailed publication on the relationship between COVID-19 and ABO blood types.

Overall, the work is written very well. The authors kept the proportions between the chapters. The manuscript is of great practical importance and is a valuable summary of the research conducted so far.

I have two main comments that I ask authors to consider:
1.the article is written in the correct language, but it should be checked by a native speaker,
2. previously published articles also summarize the relationship between ABO blood types and COVID-19 and must be cited by the authors:

Blood Rev. 2020 Dec 8;100785. doi: 10.1016/j.blre.2020.100785. Online ahead of print.

Pathogens. 2020 Jun 20;9(6):493. doi: 10.3390/pathogens9060493.

Infect Genet Evol. 2020 Oct;84:104485. doi: 10.1016/j.meegid.2020.104485. Epub 2020 Jul 30.

Author Response

We thank the reviewer his(her) interest and suggestions (in italics below):

I have two main comments that I ask authors to consider:
1.the article is written in the correct language, but it should be checked by a native speaker,

The manuscript has now been corrected by a native British English speaker.

2. previously published articles also summarize the relationship between ABO blood types and COVID-19 and must be cited by the authors:

Blood Rev. 2020 Dec 8;100785. doi: 10.1016/j.blre.2020.100785. Online ahead of print. 

Pathogens. 2020 Jun 20;9(6):493. doi: 10.3390/pathogens9060493.

Infect Genet Evol. 2020 Oct;84:104485. doi: 10.1016/j.meegid.2020.104485. Epub 2020 Jul 30.

We thank the reviewer for pointing out these references that we had missed. They have now been added. We additionally found recently published reports of meta-analyses that we also cited in the revised version of our manuscript (Pourali et al, Golinelli et al, Liu Y et al, # 90-92).

Reviewer 3 Report

The review describes the role of ABO blood group in COVID-19 risk.

The manuscript is very interesting, informative and well written.

However, I have the following suggestions:

1) The manuscript describes multiple times that the blood group O is "protective" or have "protective effect" against COVID-19. Such overstatement should be corrected using a more specific terminology such as: "blood group O is associated with lower risk of COVID-19" or "individuals with blood group O are less susceptible to COVID-19".

2) Figure 1: Genotype B/O is missing. 

3) Tables 2 and 3 should be formatted properly. 

Author Response

We thank the reviewer for his(her) interest and suggestions (in italics below):

1. The manuscript describes multiple times that the blood group O is "protective" or have "protective effect" against COVID-19. Such overstatement should be corrected using a more specific terminology such as: "blood group O is associated with lower risk of COVID-19" or "individuals with blood group O are less susceptible to COVID-19".

This has been corrected as suggested wherever observations were described. Nonetheless, we kept the word “protective” where potential mechanisms are presented in conditional tense.

2. Figure 1: Genotype B/O is missing.

We apologize for this mistake which has now been corrected. 

3. Tables 2 and 3 should be formatted properly. 

Post-submission formatting by the editorial office disrupted the original formatting of the tables. This has now been corrected.

Round 2

Reviewer 2 Report

The authors have addressed all the comments of the reviewer and revised the manuscript accordingly.